# Durable and self-hydrating tungsten carbide-based composite polymer electrolyte membrane fuel cells

Weiqing Zheng[1], Liang Wang [2], Fei Deng[3], Stephen A. Giles[1], Ajay K. Prasad[2], Suresh G. Advani[2], Yushan Yan[1] & Dionisios G. Vlachos[1]

Proton conductivity of the polymer electrolyte membranes in fuel cells dictates their performance and requires sufficient water management. Here, we report a simple, scalable method to produce well-dispersed transition metal carbide nanoparticles. We demonstrate that these, when added as an additive to the proton exchange Nafion membrane, provide significant enhancement in power density and durability over 100 hours, surpassing both the baseline Nafion and platinum-containing recast Nafion membranes. Focused ion beam/scanning electron microscope tomography reveals the key membrane degradation mechanism. Density functional theory exposes that OH• and H• radicals adsorb more strongly from solution and reactions producing OH• are significantly more endergonic on tungsten carbide than on platinum. Consequently, tungsten carbide may be a promising catalyst in self-hydrating crossover gases while retarding desorption of and capturing free radicals formed at the cathode, resulting in enhanced membrane durability.

[1] Catalysis Center for Energy Innovation and Center for Catalytic Science and Technology, Department of Chemical and Biomolecular Engineering, University of Delaware, Newark, Delaware 19716, USA. [2] Center for Fuel Cell Research, Department of Mechanical Engineering, University of Delaware, Newark, Delaware 19716, USA. [3] Department of Materials Science and Engineering, University of Delaware, Newark, Delaware 19716, USA. Weiqing Zheng and Liang Wang contributed equally to this work. Correspondence and requests for materials should be addressed to A.K.P. (email: prasad@udel.edu) or to D.G.V. (email: vlachos@udel.edu)

**F**uel cells provide a cleaner and more efficient alternative to power generation than the combustion of fossil fuels[1]. The potential of polymer electrolyte membrane fuel cells (PEMFCs) to replace the internal-combustion engine in vehicles and generate power in stationary and portable power applications has already been demonstrated[2, 3]. Yet, large-scale commercial use is currently hindered by several challenges, including the high cost of platinum (Pt) and membrane stability[4]. Proton exchange membranes (PEM), such as Nafion, currently used in fuel cells, require humidification to maintain proton conductivity using water as delivery media[5–7]. A main challenge for Nafion is how to maintain the high proton conductivity at low humidity. There have been considerable efforts to achieve this target. For example, it has been reported that adding hydrophilic materials to the membrane can improve the water retention capability[8–12]. However, this approach does not significantly enhance fuel cell performance at low humidity due to limited accessible water. Introduction of Pt nanoparticles into Nafion membranes was found to catalyze crossover $H_2$ and $O_2$ and hydrate the membrane[13–15]. Aside from $H_2O$, $H_2O_2$[16, 17] along with the formation of free radicals (OH•, and HOO•) causes chemical degradation of the membrane by attacking the C–S bond of Nafion. Moreover, adding Pt in the membrane further drives up the cost of the PEMFCs[14, 18]. Therefore, a low-cost catalyst which can more efficiently catalyze the chemical reaction of crossover $H_2$ and $O_2$ without contributing to the membrane's chemical degradation is urgently needed to improve fuel cell performance at low humidity.

Interstitial compounds of metal-C, especially the early transition metal carbides (TMCs), have been the focus of much attention following the pioneering work of Levy and Boudart[19] who showed that tungsten carbide (WC) displays Pt-like behavior in several catalytic reactions. It has been subsequently demonstrated that TMCs are effective catalysts in a number of reactions that typically utilize group VIII metals, including hydrogenation, dehydrogenation, hydrogenolysis, isomerization, and electrochemistry, with catalytic activities approaching or surpassing those of noble metals[20, 21]. However, the production of nanometer, high surface area, scalable, and importantly stable TMCs has remained a challenge.

Here, we develop a facile approach for the preparation of WC nanoparticles by a simple two-step process combining symbiotically two synthetic strategies: temperature-programmed reduction-carburization (TPRC)[22] and hydrothermal carbonization (HTC)[23]. Importantly, this method may be extended to other TMCs. We show that these WC nanoparticles, when incorporated in a Nafion membrane, are effective catalysts toward humidification and result in increased fuel cell power density with simultaneous membrane stability. Tomography of Nafion composite membranes using focused ion beam scanning electron microscopy (FIB-SEM) clearly shows significant enhancement of Nafion resistance against degradation. Density functional theory (DFT) calculations rationalize the st`ability of WC compared to Pt by exposing yet another role of WC in capturing free radicals (e.g., OH• and HOO•) from the cathode and minimizing their rate of formation on the composite membrane. Importantly, these benefits of the composite membranes are realized without increasing the PEMFC cost.

## Results

**Synthesis and characterization.** We start with the synthesis and characterization of WC nanoparticles. Conventional synthesis of WC normally entails high-temperature carburization since the incorporation of carbon atoms into a metal framework involves high activation barriers. TPRC[22] has widely been used for synthesis of high surface-area bulk-metal carbides. Unfortunately, the resulting unstable mesoporous structure restricts the applicability of this material. Alongside the utilization of TMCs, intensive efforts have been devoted to developing alternative synthesis methods to produce TMC nanoparticles with controllable size and composition. Hunt et al. reported an important three-step method to produce WC nanoparticles on a carbon support[24]. Even though the particle size of WC was below 5 nm, the long synthesis route and the large amount of template and washing agents may limit commercialization of their method. It was reported that nanosized WC can be formed by annealing tungsten precursor with polymers as carbon precursors[25, 26]. Yet, none of the reported materials showed promising catalytic activity in those applications. Recently, HTC, an old technique[23], has gained renewed interest to hydrothermally convert biomass in an aqueous solution under mild conditions to bulk, mesoporous, or nanostructured carbon materials[27–30]. Such materials can form excellent supports for WC catalysts.

Here, we leverage previous works to develop a synthesis protocol, which is simple, scalable, and may be applicable to several TMCs (Methods section). A schematic of the two-step process to prepare the WC catalyst is presented in Fig. 1. HTC of glucose leads to the formation of solid carbon spheres through dehydration and polymerization reactions, which can encapsulate transition metal precursor nanoparticles. After filtration and washing, the resulting composite particles were subjected to a TPRC process, wherein the carbon spheres serve as "spacers" to prevent sintering of WC nanoparticles.

The morphology of the resulting WC nanoparticles is shown in Fig. 2, Supplementary Figs. 1 and 2. The as-prepared catalyst carbon support has a smooth spherical structure with a diameter of 3–5 μm containing well-dispersed WC nanoparticles on its surface. Using dual-beam SEM, we examine the cross-sectional morphology of individual carbon spheres[31]. As shown in

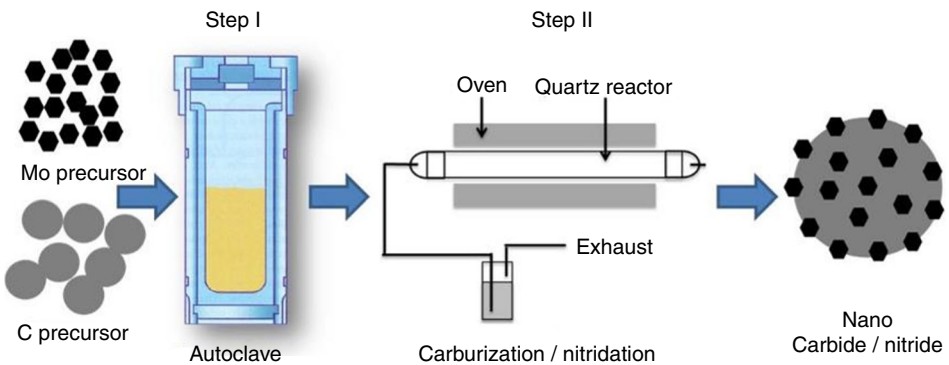

**Fig. 1** Schematic of a two-step method for the synthesis of transition metal carbide nanoparticles dispersed on a carbon material

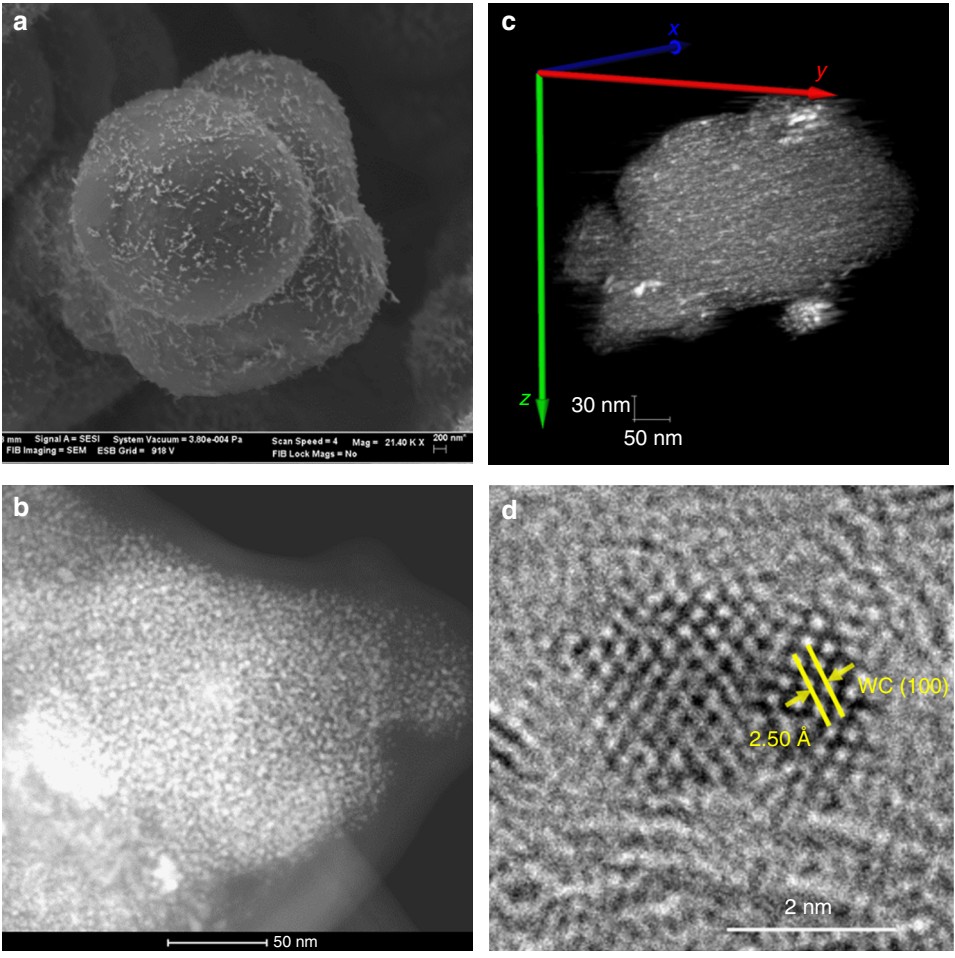

**Fig. 2** Electron microscopy images of as-prepared WC nanoparticle catalyst. **a** SEM image of a representative carbon sphere with well-dispersed WC nanoparticles. **b** HAADF-STEM image. **c** STEM tomography of the resulting WC nanoparticles dispersed on carbon. 3D volume rendering from the reconstruction of the aligned tilt series images recorded by tilting the sample from −65° to +55° with 1° increment. **d** HRTEM image of a representative WC nanoparticle with lattice index measurement corresponding to the WC(100) surface

Supplementary Fig. 1, the WC particles are located on the surface of carbon spheres, although W signal is found across the entire sphere (see energy-dispersive X-ray spectroscopy (EDX) mapping). The high-angle annular dark field (HAADF) image over relatively large scales (Fig. 2b) and the three-dimensional (3D) rendering of the reconstructed tilt series (Fig. 2c) show that the WC nanoparticles are uniformly dispersed on the carbon sphere surface with a narrow size distribution (around 3–5 nm). The representative bright field transmission electron microscopy (TEM) images (Supplementary Fig. 2) also confirm the presence of well-structured WC nanoparticles dispersed on carbon. The high resolution TEM (HRTEM) image (Fig. 2d) reveals the hexagonal close-packed $\alpha$-WC structure, which was confirmed by lattice index measurements. The minor intensities of $\alpha$-WC diffraction patterns shown in Supplementary Fig. 3b indicate a small crystalline size. The surface properties of the WC nanoparticles were further analyzed by X-ray photoelectron spectroscopy (XPS) and summarized in Supplementary Fig. 4. The W4$f$ doublets at 31.6 and 33.7 eV, the carbidic carbon peak at 282.6 eV, the minor intensity of oxygen features, and the high density at the Fermi level reveal the WC dominate the surface of WC nanoparticles. Based on this structural evidence, we can confirm that our two-step preparation method successfully produces WC nanoparticles smaller than 5 nm supported on, and separated by, solid carbon material, which prevents nanoparticle sintering and results in a high surface area material.

Importantly, the method can be extended to other TMCs nanoparticles.

**Fuel cell performance**. Fuel cell performance was carried out with the baseline recast Nafion membrane, a Nafion membrane with our 5 wt.% WC catalyst, a Nafion membrane with 5 wt.% Pt black catalyst, and a Nafion membrane with 5 wt.% commercial WC particles (average particle size of around 55 nm; see SEM image of commercial WC in Supplementary Fig. 7). All tests were conducted at a cell temperature of 70 °C. $H_2$ and $O_2$ flow rates were 200 and 400 ml/min, respectively. The gas supply lines were maintained at 75 °C to prevent condensation of water vapor. The temperature of the humidifier was controlled at 70, 55, 41, and 14 °C (cooling water temperature) to achieve relative humidity (RH) of 100, 50, 25, and 5%, respectively. All results are summarized in Supplementary Fig. 9.

Figure 3a shows that humidity has a strong effect on the recast baseline Nafion membrane, which lost most of the power at 5% RH and underscores the main reason of applying external humidity at the expense of increasing mass, size, and cost of PEMFCs[32]. Upon introducing catalysts in the membrane, the fuel cell performance is improved due to the water generation from crossover $H_2/O_2$ to humidify the bulk Nafion membrane and thereby increase efficiency. Clearly, our composite membranes significantly improve the proton conductivity at low RH,

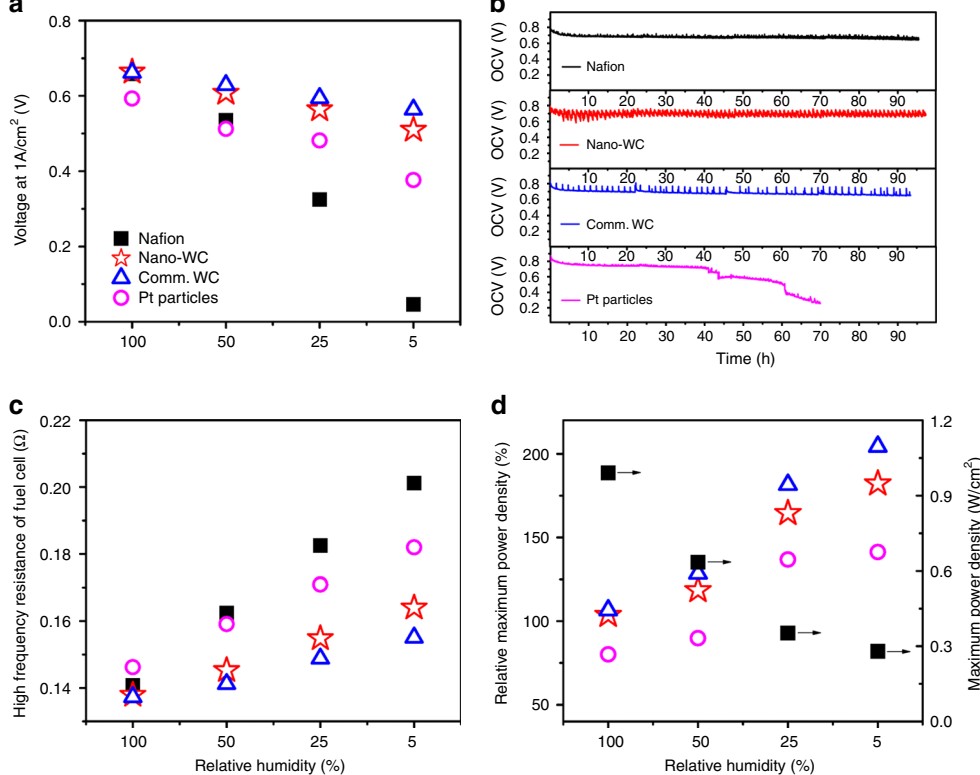

**Fig. 3** Fuel cells performance of recast Nafion and composite membranes. **a** Fuel cell performance at 1 A/cm$^2$ condition under 5, 25, 50, and 100% relative humidity. **b** Proton conductivity of the fuel cell with recast Nafion membrane MEA and composite membranes MEAs measured by two-probe electrochemical impedance spectroscopy. **c** 100 h accelerated fuel cell durability tests. **d** Relative maximum power density ($P_c/P_N \times 100$, where $P_c$ and $P_N$ are the maximum power densities of the composite membranes and pristine Nafion membrane, respectively) measured at different relative humidity. *Closed squares* represent the pristine Nafion membrane, *open circles*, *stars*, and *triangles* represent the composite membranes with 5wt. % commercial WC, nano-WC, and Pt, respectively

as shown in Fig. 3b. At high RH, the composite membranes show minor improvement of fuel cell performance, likely due to the high proton conductivity owing to the external water. Upon introducing WC nanoparticles in the membrane, the fuel cell's performance improves by about 20% (at 50% RH) and 80% (at 5% RH) compared to the baseline Nafion membrane, and approaches that of the recast membrane containing 5 wt.% Pt. Although the commercial WC composite membrane surpasses the pristine Nafion membrane at low humidity, the inter-particle pores inside bulk WC clusters limit the access of Nafion precursor and the transport of protons through the bulk WC's pores at higher RH. This problem is surpassed by WC nanoparticles (nano-WC) inside the Nafion membrane, because they provide much higher density of active sites and prevent transport limitations by being located on the surface of non-porous carbon spheres (Fig. 2). Even though the Pt composite membrane has the highest peak power density, further increase in power density may be possible by optimizing the WC loading.

Besides the improvement of fuel cell performance at low humidity, the durability of the membrane is a major impediment preventing the broad application of fuel cells[33]. The Department of Energy's (DOE) durability target for transportation fuel cells is 5000 h (equivalent to 150 000 miles of driving) under realistic operating conditions from the current 3900 h. Improving stability of the FC materials is needed to facilitate commercialization. It is generally believed that the perfluorosulfonic acid membrane undergoes attack by free radicals (e.g., OH•, OOH•) resulting from hydrogen peroxide ($H_2O_2$) forming at the cathode by the two-electron reduction of $O_2$ due to cross-leakage of $O_2$ gas.

Chemical attack by these reactive radicals thins out the membrane and exacerbates the formation of mechanical defects[34]. Therefore, it is important to assess the durability of the WC composite membrane.

In order to evaluate the durability of the self-hydrating composite membrane, we carried out a series of accelerated fuel cell operating tests according to DOE protocol at 90 °C and 35% RH under open circuit voltage conditions for 100 h (Fig. 3c)[35]. High temperature and low humidity have been recognized as the most effective conditions for fuel cell degradation[36], and the OCV test, which results in high gas permeability and more free radicals, is believed to accelerate the chemical degradation of the membrane[37]. As shown in Fig. 3d, a slight decline of voltage was observed for the pure recast Nafion membrane after 100 h with degradation rate $0.285 \pm 0.003$ mV/h (Supplementary Fig. 10a). The degradation rate of Nafion membrane with commercial WC is $0.625 \pm 0.005$ mV/h slightly faster than the recast Nafion membrane (Supplementary Fig. 10d), probably due to the large particle size of WC, which causes higher initial gas crossover (Supplementary Fig. 10). The Pt composite membrane showed a slow decreasing voltage after 24 h (degradation rate $1.38 \pm 0.01$ mV/h, Supplementary Fig. 10c) followed by an accelerated voltage drop (degradation rate of $6.09 \pm 0.04$ mV/h and $14.3 \pm 0.14$ mV/h, respectively). In contrast, the composite membrane containing nano-WC showed no discernible decline of voltage for 100 h. As shown in Supplementary Fig. 10b, a slow $0.05 \pm 0.008$ mV/h degradation rate was observed over the composite membrane containing nano-WC, which is 1/5 of the recast Nafion's rate. A minor loading of nonprecious catalyst can profoundly enhance the PEMFC performance and durability.

**Fig. 4** Investigation of fresh and used Nafion membranes by FIB-SEM tomography. The *white* regions correspond to the catalyst (Pt or WC nanoparticles) embedded in the Nafion membrane, while voids within the Nafion membrane are depicted in *red*. **a** A representative region of PEM sample during FIB-SEM tomographic investigation. The focused ion beam removes a 12 nm-thick layer of the membrane during each pass, while an electron beam records its SEM image. **b**, **c** 3D reconstruction of fresh Pt/Nafion and WC/Nafion membranes, respectively. **d–f** 3D reconstruction of used Nafion, Pt/Nafion and WC/Nafion membranes, respectively, after 100 h of fuel cell operation

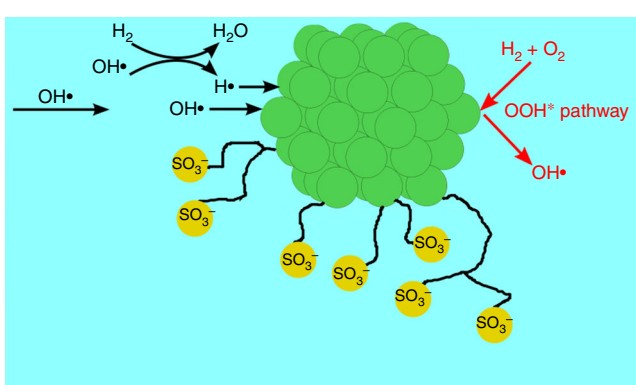

**Fig. 5** Schematic of the interaction of the embedded nanoparticles with radicals in solution. *Green circles* represent atoms of the embedded nanoparticle, and *yellow circles* represent sulfonic groups of the Nafion that are susceptible to degradation

The measured PEMFC membranes were further analyzed using cross-sectional SEM images (Supplementary Fig. 12) and FIB-SEM to reveal the detailed 3D microstructure. Figure 4a shows a PEM sample during FIB-SEM tomography in which the milled region is identified. Figure 4b–f present the 3D morphology of fresh and used membranes reconstructed from

the FIB milled cube. Both recast Nafion (Fig. 4d) and our 5 wt.% WC/Nafion (Fig. 4f) membranes showed in-plane pinholes throughout the membrane after 100 h of the accelerated durability test. Because the durability test was conducted at 35% RH in both the anode and cathode streams, diffusion-induced water flux through the membrane thickness is not expected. Therefore, pinholes form along the in-plane direction, the primary direction of water flux within the membrane (from inlet to outlet). Figure 4b, f reveal almost identical pinhole morphologies; the voids are small and highly aligned in the in-plane direction. Importantly, the voids in Fig. 4f do not show any preferential clustering adjacent to the WC catalyst nanoparticles. In contrast, in the Pt-based membrane, the pinholes are large and highly clustered around the Pt catalyst (Fig. 4e). This finding hints to yet another role of Pt; aside from hydrating the membrane (a beneficial effect), it produces radicals that locally degrade the membrane (an undesired effect).

The pinhole void fractions of the used membranes can be estimated using 3D tomographic analysis, as summarized in Fig. 4. The results are in agreement with the gas crossover quantifications measured during stability tests. The smaller void fraction in the WC/Nafion sample after the durability test is consistent with its smaller gas crossover shown in Supplementary Fig. 13. From the FIB-SEM and gas crossover measurements, it may be concluded that the nano-WC catalyst does not cause damage to the Nafion polymer. Instead, it enhances the stability

of the Nafion polymer. In contrast, the Pt catalyst causes severe damage of the Nafion membrane with defects forming around the Pt black particles, possibly due to the high concentration of free radicals generated from the catalyst during the crossover $H_2/O_2$ reaction. Hence, the WC catalyst is able to improve fuel cell performance by enhancing the membrane's self-hydration ability, without the harmful effect on stability, and it does so at low cost.

**First-principles calculations.** With the aim of revealing the intrinsic mechanisms of the reactions occurring on the embedded nanoparticles and rationalize the enhanced stability of the membrane, we have conducted DFT calculations. We assess different mechanisms that can influence the concentration of OH• and H• radicals in solution, which are believed to retard Nafion via removing $SO_3^-$ groups. A schematic of the reaction pathways is shown in Fig. 5. First, OH• produced from the Pt electrocatalyst cathode may directly be captured on the nanoparticle surface via adsorption,

$$OH\bullet + * \rightarrow OH^*, \qquad (1)$$

where * represents a vacant site on the nanoparticle. In addition, H• may be produced as a result of OH• donating its oxygen to $H_2$. H• may then be captured by the nanoparticle through a similar adsorption step:

$$OH\bullet + H_2 \rightarrow H\bullet + H_2O, \qquad (2)$$

$$H\bullet + * \rightarrow H^*. \qquad (3)$$

The production of H• from OH• has been reported to be exothermic of a low energy barrier[38]. Although H• has a short lifetime, H• produced in solution can still affect the stability of Nafion[39]. Aside from adsorbing OH• and H• from solution, OH• is produced on the membrane catalyst during $H_2$ oxidation and can desorb to solution. The potential for this chemistry on nanoparticles is of particular concern since in situ production of OH• and H• could lead to major deterioration of Nafion stability. The reaction mechanism of OH• production considered herein on both WC and Pt is analogous to that reported in the literature[38].

DFT calculations reveal that WC binds OH• and H• more strongly than Pt (adsorption is more exergonic on the WC by 2.29 and 0.63 eV, respectively) and is thus facilitated by WC. As a result, WC more efficiently captures radical species from solution produced on the Pt electrode. Investigation of the ability of the Pt and WC nanoparticles to in situ produce OH• clearly shows that production of OH• is highly unfavorable relative to the oxidation of the WC surface by $O^*$. The potential energy diagram in Supplementary Fig. 14a of the OH• formation mechanism on Pt (111) shows that the most favorable pathway for OH• formation is the dissociation of HOOH* to form adsorbed $OH^*$ and an OH• radical in solution, with a reaction free energy of +0.56 eV. In contrast, the potential energy diagram for the OH•-formation mechanism on WC(100) (Supplementary Fig. 14b) reveals that the most favorable pathway for OH• formation is the dissociation of OOH* to form $O^*$ and an OH• radical in solution, with a reaction free energy of +4.01 eV. As a result, a significantly larger thermodynamic barrier exists to form OH• radicals on WC(100) than on Pt(111). Therefore, DFT calculations indicate that incorporation of nano-WC can benefit the stability of the Nafion structure first by adsorbing radical species already in solution released from the cathode, and second by being relatively inactive towards OH• production.

## Discussion

In summary, we report here a facile and scalable two-step process for large-scale production of WC nanoparticles of a narrow size distribution (3–5 nm) as a result of the surrounding carbon preventing sintering. This protocol may be extended to other early TMCs, such as molybdenum carbide nanoparticles (Supplementary Fig. 8). Importantly, it may be possible to produce other interstitial compound nanoparticles (e.g., transition metal nitrides and sulfides) to replace noble metal catalysts in a wide range of chemical reactions given that the formation paths for these materials typically parallel those of carbides. Our preliminary experiments of using WC nanoparticles as electrodes in fuel cell tests (Supplementary Fig. 15) demonstrate their effective catalytic performance.

WC nanoparticles incorporated into a recast Nafion membrane demonstrates significant improvement of the PEMFC power density compared with the baseline Nafion membrane by ~80% at 5% (low) RH. Similar loadings of commercial WC do not increase performance due to its low surface area. Based on both theoretical and experimental results, the WC composite membrane catalyzes the reaction of crossover $H_2/O_2$ and provides sufficient water management within the membrane, improving the proton conductivity. Tomographic studies using FIB-SEM indicates large pinholes forming around Pt nanoparticles and provides support so that the membrane/Pt catalyst facilitates the production of radicals that leach away the $SO_3^-$ groups of Nafion. Superior to Pt, incorporated WC nanoparticles in the membrane results in considerable durability of the membrane by eliminating formation of large pinholes. DFT calculations indicate a dual role of WC in enhancing membrane stability by simultaneously capturing free radicals generated at the cathode and by inhibiting the formation of $H_2O_2$ and free radicals from crossover $H_2/O_2$. The composite membrane may bring fuel cells a step closer to widespread commercialization.

## Methods

**Materials synthesis.** An aliquot of 35 ml aqueous solution of ammonium metatungstate hydrate (AMT, Sigma-Aldrich) and D(+)-glucose (Sigma-Aldrich) was added to a 50 ml non-stirred Teflon-lined autoclave, which was placed in a temperature-programmed muffle furnace and heated to 200 °C for 2 h. After filtration, the solid paste was washed with 500 ml deionized water four times, and dried overnight at 110 °C. The TPRC process was carried out in a quartz reactor tube at 750 °C. An aliquot of 20 ml of 5% Nafion solution (D-521, ≥0.92 meq/g, Alfa Aesar) was dried at 60 °C to vaporize the solvent. The Nafion resin was then dissolved in dimethylacetamide (DMAC) to form Nafion/DMAC solution and 47 mg of the previously prepared WC NPs was added to it. The mixture was sonicated for at least 2 h to mix with Nafion. Then the Nano-WC/Nafion solution was poured onto a glass plate and heated in an air oven at 120 °C for 4 h, and in a vacuum oven at 150 °C for 2 h. The cured membrane of thickness 50 μm was then lifted off the glass plate and immersed in 0.5 M sulfuric acid for 2 h, and rinsed with DI water. Finally, the composite membrane was dried and hot-pressed between gas diffusion electrodes with 0.3 mg/cm$^2$ Pt loading at 130 °C for 2 min to fabricate the membrane electrode assembly (MEA). The MEA's performance was then tested in a 5 cm$^2$ fuel cell. Baseline Pt/Nafion membranes were also prepared using the same procedure. In addition, two more composite membranes, one incorporating commercial Pt black and the other WC nanopowder, were also prepared for comparison.

**Fuel cell measurements.** The polarization $I$–$V$ evaluation of the fuel cell was conducted and controlled by a fuel cell test station from Arbin Instruments. The $H_2$ and $O_2$ humidifiers were maintained at 70, 55, 41, and 14 °C while the fuel cell temperature was set to 70 °C such that the RH of the inlet gases was 100, 50, 25, and 5%, respectively. Hydrogen fuel and oxygen were fed in co-flow to the fuel cell. $H_2$ and $O_2$ flow rates were 200 and 400 ml/min, respectively. The fuel cell tests were conducted at ambient pressure. The fuel cell was conditioned for 8 h at a current density of 1 A/cm$^2$ before collecting performance data. For each humidify, the fuel cell was discharged at 0.5 A/cm$^2$ for 1 h and held at OCV for 10 min before $I$–$V$ evaluation. Two-probe electrochemical impedance spectroscopy measurements were carried out using a VersaSTAT 3 potentiostat (PrincetonApplied Research) to fuel cells with VersaStudio data acquisition software in the frequency range of 10 000 Hz to 0.1 Hz. Impedance data were fit to a typical Randle's circuit using the ZView plotting software (ScribnerAssociates).

All experiments were carried out at cell temperature of 70 °C, with 100 ml/min $H_2$ and 200 ml/min $O_2$, respectively.

**Materials characterizations.** X-ray diffraction patterns of the catalyst and the catalyst-Nafion composite membranes were performed with a Bruker D8 Discover diffractometer using CuKα radiation ($\lambda K = 1.540$ Å). Measurements were taken over the range of 5° < 2θ < 95° with a step size of 0.05° and a count time of 1 s at each point. The HRTEM images were obtained with a JEOL JEM-2010F transmission electron microscope equipped with a field emission gun (FEG) emitter. The HAADF images and 3D tomography characterization were acquired on a FEI Talos F200C microscope equipped with a FEG emitter. A thermo-fisher K-alpha+X-ray photoelectron spectrometer equipped with a monochromatic Al-Kα X-ray source (400 μm analysis spot size) was used for XPS analysis. The surface morphology and 3D structure of the samples were investigated with a cross-beam SEM with a Ga+ion source FIB (Auriga-60, ZEISS). The specimen was mounted on a cross-section sample holder facing the ion column, while the SEM images were recorded from a side view at an angle of 54°. The Ga+ion beam used to mill the sample was operated with an energy of 30 kV, and a current of 600 pA. A 1 μm-thick Pt thin-film was deposited in situ on the sample's surface in order to prevent damage to the surrounding regions during milling. The size of the 3D milled volume was 20 μm wide×20 μm long×12 μm deep. A total of 1000 2D slices was collected at a depth-resolution of 12 nm. Each 2D slice was imaged at 2058 × 2058 pixels with an e-beam energy of 3 kV and an In-lens detector for high-contrast imaging. The 3D reconstruction of the sample was performed with the analytical software Avizo 7 (FEI Company).

**Theoretical calculations.** Periodic DFT calculations were performed using the Vienna ab-initio software package (VASP, version 5.3.2)[40]. The Perdew–Burke–Ernzerhof exchange-correlation functional[41] was used to approximate the exchange-correlation energy. The core electrons were represented with the projector-augmented wavefunction method[42, 43] and a plane-wave cutoff of 400 eV was used for the valence electrons. The Methfessel–Paxton method[44] of electron smearing was used with a smearing width of 0.1 eV. All geometry optimizations were performed using the conjugate gradient algorithm[45] as implemented in VASP. The forces and energies were converged to 0.05 eV Å$^{-1}$ and $10^{-4}$ eV, respectively. All calculations were performed spin-polarized. To model the surface sites of the Pt and WC nanoparticles embedded within the Nafion structure, the surface energies of various planes of the Pt and WC bulk structures were calculated to determine the most relevant experimental surfaces to model. The Pt (111) plane and the WC (100) plane were found to be the lowest energy surfaces (Supplementary Table 3). For all surface calculations, a 3 × 3 periodic unit cell was used. In the z-direction, a vacuum layer of 15 Å was included. A 3 × 3 × 1 Monkhorst-Pack k-point sampling of the Brillouin-zone was implemented[46]. Zero-point energy and temperature corrections, performed at 298.15 K, were taken from previously published data[47]. Higher accuracy calculations (5 × 5 × 1 k-point sampling and a 500 eV plane-wave cutoff) were also performed for H• and OH• adsorption on Pt (111) and WC (100). In all cases, the adsorption energy changed by < 0.05 eV.

**Data availability.** The data supporting the findings of this study are available in the Supplementary Information and raw data from the corresponding authors upon request.

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

## Acknowledgements

D.G.V., Y.Y., W.Z., and S.A.G. acknowledge support from the Catalysis Center for Energy Innovation, an Energy Frontier Research Center funded by the U.S. DOE, Office of Science, Office of Basic Energy Sciences under Award No. DE-SC0001004 for the WC nanoparticle synthesis, the materials characterization, and the DFT calculations as part of synthetic control of materials for energy. A.K.P., S.G.A., and L.W. acknowledge support from the University of Delaware's Fuel Cell Bus Program for conducting the fuel cell tests. This program is funded by the Federal Transit Administration at the Center for Fuel Cell Research at the University of Delaware.

## Author contributions

W.Z. implemented the synthesis of WC nanoparticles and performed the characterization of materials. L.W. prepared the composite membranes and investigated fuel cell performance. S.A.G. performed the DFT calculations. F.D. carried out the FIB-SEM analysis. A.K.P., S.G.A., Y.Y., and D.G.V. conceived the problem and participated in the paper writing.

## Additional information

**Competing interests:** The authors declare no competing financial interests.

