## [Peer Review File · Nature Communications]

Reviewers' comments:

Reviewer #1 (Remarks to the Author):

The manuscript describes the use of tungsten carbide mixed membrane to improve the performance and durability of the fuel cell. I do not think that the novelty of the manuscript meet the standard of the Journal.

Comments:

- 1, The authors did not cite proper papers, particular, form active groups.
- 2, The authors concluded that "the WC composite membrane catalyze the reaction of crossover H₂/O₂ and provides sufficient water management within the membrane", But, I could not find the evidence to prove it.
- 3, The mechanism of H[•] formation and the lifetime did not concern. All the processes that the authors described could be danger since the lifetime of H[•] is only 10⁻⁹ s. How it can be existed from the free-standing state to adsorb on the surface of WC or solution?

Reviewer #2 (Remarks to the Author):

The authors present a method to synthesize transition metal carbide nanoparticles with tunable size. These particles are used in a proton exchange Nafion assembly to deliver a 20% enhancement in power density and durability. Combined characterization and DFT studies were performed to explain the observed behavior. There are several inconsistencies in the manuscript that will be described in more detail below; overall, the main finding of the paper does not appear to provide a significant improvement over the baseline. In addition, some of the interpretations of the nanoparticle structure appear to have inconsistencies as well. Therefore I cannot recommend publication in Nature Communications. My comments are the following:

1. Materials Characterization

The authors claim that

a) "our simple two-step preparation method successfully produces metal-terminated WC nanoparticles of less than 5 nm supported on, and separated by, solid carbon material, which prevents nanoparticle sintering and results in a high surface area material. Importantly, the method can be extended to the synthesis of other TMC nanoparticles."

b) "In summary, we report here a facile and scalable two-step process for large-scale production of tunable in size WC nanoparticles."

and c) "Importantly, we envisage applicability of our method to produce other interstitial compound nanoparticles (e.g., transition metal nitrides and phosphides) that could replace noble metal catalysts in a wide range of chemical reactions."

These broad claims are based on data for a single WC materials using only SEM/TEM imaging with an EDX map in Figure 2 and Figure S1 along with a PXRD spectrum in Figure S2b. Unfortunately, this single synthetic example and the techniques used are inadequate to confirm metal termination or the broad applicability of the method. The authors should provide at least XPS data, better PXRD patterns, or XAS data to corroborate claims. More specifically, a metal-terminated WC surface will not exhibit a passivation layer with WO_x. Otherwise, it should be denoted as O-terminated. Furthermore, a carbidic C1s shift should be detectable at the surface and it must be comparable or lower in intensity to the adventitious carbon signal. Without this important information, it is unreliable to model a W-terminated material with DFT.

-If the particles are separated by carbon, it is highly likely they are coated in graphitic coke.

Ultimately, it is unclear how the materials are different from those obtained by mixing W precursors with a carbon matrix and heating to carburize the metal with carbon originating from the support, which result in WC nanoparticles encapsulated in graphitic coke. The authors should refer to Z. Yan, M. Cai, and P.K. Shen, *Scientific Reports*, 2013, 50, 1646-1653. or R. Ganesan and J.S. Lee, *Angew. Chem. Int. Ed.* 2005, 44, 6557-6560.

-The bright white spots on the surfaces of the carbon spheres in Figure 2a are called "well dispersed WC nanoparticles." Based on the scale bar, these particles are at least 20-30 nm in size. Figure S1 shows a cutaway.

- The authors claim that the method provides size tunability, but no examples are shown where different particles sizes are achieved. Specifically, multiple syntheses have to be performed and the PSDs have to be compared. They must be narrow and statistically different.

-Figure 1 shows nitridations being performed. No data on nitrides is shown. This must be removed. The "Nano Carbide/Nitride" depiction is also inaccurate based on Figure S1. The WC particles are inside the carbon sphere.

-The authors claim to synthesize "hexagonal close packed β -WC". I believe, hexagonal close packed WC is typically referred to as just WC or α -WC. Cubic WC synthesized in a face centered cubic lattice is referred to as β -WC or WC1-x. In Figure 2d, the authors claim to observe the WC(100) plane. β -WC does not have this plane. In Figure S2b, the authors index WC nanoparticles diffractogram to α -WC; however the WC nanoparticle diffractograms cannot be indexed as the intensities are too low.

2. Membrane Testing and Characterization:

The title of the paper includes "Self-hydrating." In the second paragraph of the introduction, the paper is motivated by using nanoparticles incorporated in the membrane to "self-hydrate" the membrane via water produced in the cathode. I agree with the authors that not having to have a complex humidification system would significantly simplify the balance-of-plant. However, if this is a main motivation for the paper, then why was the fuel cell testing performed with both H₂ and O₂ humidifiers maintained at 55 °C? If the authors wish to claim self-hydration and include it in the title, the studies with WC and Pt in the membrane have to be shown without the humidification systems. Otherwise, it is hard to justify the presence of "Self-hydrating" in the title.

Figure 3a shows that there is no improvement of fuel cell performance with 5wt% commercial WC, likely due to the small surface area of WC leading to limited oxidation of crossover H₂. The commercial catalysts are typically prepared through ball-milling, which could lead to some metal termination. Based on Figure S3, the surface area is likely not that low. The authors should run N₂ adsorption/desorption isotherms on the materials and measure the surface area. Based on the authors' hypothesis throughout the paper and the DFT calculations, this commercial WC/Nafion control should have exhibited some improvement, but instead it is actually worse than just Nafion. The data from this control do not support one of the main conclusions of this paper that metal-terminated WC is catalyzing the formation of H₂O from crossover of H₂ to hydrate the membrane.

3. DFT calculations

The DFT calculations are performed on idealized, metal-terminated WC. While it is understandable that idealized surfaces must be used in DFT calculations, this assumption likely does not hold in the context of the conditions that WC is being subjected to. Specifically, the conditions that the authors are exposing the WC materials to means that they will be passivated in an oxide layer. Not only will WC be passivated, it will likely be thickly passivated. The surface is at least an oxycarbide and more likely is actually similar to WO₃. As a result, it is difficult for me to gain any information from the DFT calculations to understand the observed improvements in membrane stability.

Why was such a low k-point mesh and a low cutoff energy used for the calculations? To model the WC surface under operating conditions, these calculations should be done with at least a 5x5x1 kpoint mesh and a cutoff energy of at least 500 eV.

Reviewer #3 (Remarks to the Author):

Several diatomic transition metal carbides or oxides with n valence electrons have similar spectroscopic and chemical properties as transition metal atoms with n d-electrons (Castleman). WC thus simulates Pt and has been investigated as replacement for Pt as a catalyst. In this sense, the present work uses nano-size WC as an additive to Nafion in place of Pt and studies its influence on the power density and the stability of the Nafion fuel cell membrane. An improved behaviour is ascribed to the self-hydrating property due to the conversion of the H_2 and O_2 cross-over gases to water inside the membrane, and DFT calculations find that the $H\cdot$ and $HO\cdot$ radicals adsorb more strongly on WC than on Pt so that they are less available for damaging radical reactions on the polymer electrolyte. Furthermore, a tomographic technique reveals interesting pattern of degradation.

This is an interesting topic, and the work combines a variation of different methods. The manuscript reads quite well, and the interpretations sound plausible. However, if one goes into details one gets the impression of a somewhat abridged study with quick conclusions which may be correct but lack certainty in the absence of cross-checks or comparison with literature:

1. Conversion of cross-over gases in the membrane at the added catalyst particles is concluded from the Ohmic region of the polarization curves, but without quoting the membrane resistances derived from the slopes. Perhaps more commonly, the reductions of the open circuit voltages are judged for the same purpose. These, however, seem inconsistent with the suggested interpretation (Figure 3a).
2. Durability tests are conducted over durations of 100 h OCV. This may be partly justified by the external humidification of the gases which is only 50%, but it is very short compared with normally expected lifetimes of 5000 h under running conditions. What bothers more is the low number of experiments. Is it really enough for a reliable conclusion considering that the breakdown that is seen for 5 wt.% Pt/Nafion normally sets in at random time, depending on the actual pinhole situation.
3. As a degradation mechanism it is suggested without any reference that $H\cdot$ and $HO\cdot$ radicals in solution react with Nafion via removing SO_3^- groups. I do not know how $H\cdot$ radicals should form in solution since they are strongly bound to the Pt catalyst (supplementary information). Furthermore, it is known in literature from studies with model compounds that SO_3^- groups are much more inert towards attack of $HO\cdot$ radicals than other sites of Nafion [Dreizler and Roduner, Fuel Cells, 12 (2012) 132-140]. Furthermore, with a standard reduction potential of 1.77 V, $HO\cdot$ radicals are strongly oxidizing species [Koppenol and Liebman, J. Phys. Chem. 88 (1984) 99.], an aspect which is unfortunately not normally considered in mechanistic work on membrane degradation.
4. VASP calculations were limited to the Pt(111) and the WC(100) facets although it is known that oxygen reduction is quite different for the various low-index Pt surfaces, and no comparison is made with literature data on very detailed previous work [Panchenko et al. J. Electrochem. Soc. 151 (2004) A2016; and J.K. Norskov et al. J. Phys. Chem. B, 108 (2004) 17886.].

Overall, this work reminds more of a conference proceeding of work in progress, but it does not meet the standard expected for a Nature Comms. Paper.

Review #1

The manuscript describes the use of tungsten carbide mixed membrane to improve the performance and durability of the fuel cell. I do not think that the novelty of the manuscript meet the standard of the Journal.

Comments:

1, The authors did not cite proper papers, particular, form active groups.

We appreciate the reviewer's comment on this issue. We revised the introduction to cite all relevant papers in our manuscript. We would like to highlight the novelties of our work here:

1. We report a facile, scalable, two-step method for the production of tungsten carbide (WC) nanoparticles which differs from previous strategies reported in literature. Recent efforts have been devoted to producing nanoparticles but these approaches are too complex to be economically viable, or result in materials that are either not stable or that serve as a support rather than as the active component. Our synthetic protocol overcomes these hurdles and offers the potential of wide use of transition metal carbides (TMCs) as **catalysts**. To demonstrate the broad applicability of our synthetic approach we provide in Supplement for reviewers only results (supplementary discussion in SI together with Figure R1) demonstrating the formation of Mo₂C.
2. We introduce WC nanoparticles in a recast Nafion membrane and demonstrate significant improvement of the PEMFC performance that approaches the power density of Pt while, unlike Pt, gives durable performance over long periods of time and much lower cost. While revising our manuscript, Park *et al.* **recently** demonstrated the crucial importance of water management in PEMFCs with the self-humidifying concept in the *Nature* journal (C.H. Park *et al.* *Nature*, 532, **2016**, 480-483). We believe that carefully preparing a composite membrane can balance humidity, inhibit membrane degradation, and maintain high performance for long times. Importantly, our well-controlled nano-WC provides both durability and water management at a much lower cost.

We have now included all literature that is pertinent to the novel aspects listed above.

2, The authors concluded that "the WC composite membrane catalyze the reaction of crossover H2/O2 and provides sufficient water management within the membrane", But, I could not find the evidence to prove it.

This is an important point. In order to address the reviewer's comment, we have performed additional experiments under different conditions as described below:

Fuel cell performance of membranes containing Pt, commercial WC, and nano-WC catalysts and baseline recast Nafion membranes were tested and compared under 100%, 50%, 25% and 5% relative humidity (RH), (Figure S5 in SI). Durability of the Nafion membrane with commercial WC was added (Figure 3c). Proton conductivity data of different membranes under various RH conditions was also added to the manuscript (Figure 3b). Gas crossover during the durability test

of the Nafion membrane with commercial WC was added (Figure S6 in SI). Additional text was added to the discussion section.

By comparing the fuel cell performance with Pt, commercial WC, nano-WC and recast Nafion membranes at various humidities, we clearly demonstrate that the membranes containing the nano-WC catalysts show improved performance and better proton conductivity (Figure 3 and Figure S6).

3, The mechanism of H• formation and the lifetime did not concern. All the processes that the authors described could be danger since the lifetime of H• is only 10-9 s. How it can be existed from the free-standing state to adsorb on the surface of WC or solution?

We apologize for this confusion about the H• which was considered together with OH• and OOH• radicals when we conducted density functional theory (DFT) calculations to rationalize the mechanisms on both the Pt and the WC surface. The general consensus is that OH• and OOH• radicals forming at the cathode desorb into solution that degrade the Nafion membrane, as identified in PEMFCs (both membrane and electrodes) using electro spin resonance (ESR). (M. Danilczuk *et al. J. Phys. Chem. B*, **2009**, *113*, 8031; S. Schlick *et al. Molecular Physics*, **2013**, *111*, 2738.) However, there has also been evidence of H• leading to Nafion degradation (Ghassemzadeh, L. *et al. J. Phys. Chem. C* **2010**, *114*, 14635-14645; Yu, T. H. *et al. J. Am. Chem. Soc.* **2011**, *133*, 19857-19863). Specifically, Ghassemzadeh *et al.* found that the CF backbone carbon can be attacked by H•, producing HF in the process. Comparing to other degradation pathways considered, this mechanism was confirmed to have a relatively low barrier (1.00 eV) in a subsequent DFT study (Yu, T. H. *et al. J. Am. Chem. Soc.* **2011**, *133*, 19857-19863). As shown in Reaction 2 in the manuscript, H• can be produced by OH• reacting with crossover H₂. This reaction has been previously reported to be exothermic (reaction free energy of -0.59 eV) with a very low energy barrier (0.04 eV) (Yu, T. H. *et al. J. Am. Chem. Soc.* **2011**, *133*, 19857-19863). Because the production of H• from OH• in solution is known to be facile, attack of the Nafion by H• is a possibility which cannot be excluded. Therefore, we believe that the ability of Pt and WC to capture H• must be studied in our first-principles investigation to be complete. We have added a note to the manuscript which clarifies the necessity of studying H• adsorption on the nanoparticle.

Review #2

The authors present a method to synthesize transition metal carbide nanoparticles with tunable size. These particles are used in a proton exchange Nafion assembly to deliver a 20% enhancement in power density and durability. Combined characterization and DFT studies were performed to explain the observed behavior. There are several inconsistencies in the manuscript that will be described in more detail below; overall, the main finding of the paper does not appear to provide a significant improvement over the baseline. In addition, some of the interpretations of the nanoparticle structure appear to have inconsistencies as well. Therefore I cannot recommend publication in Nature Communications. My comments are the following:

We completely understand the reviewer's concerns. We have included further experimental evidence and discussion (described in detail below) to support our claims regarding materials synthesis and PEMFC performance and rephrased statements to avoid inconsistencies in interpretation.

1. Materials Characterization

The authors claim that

a) "our simple two-step preparation method successfully produces metal-terminated WC nanoparticles of less than 5 nm supported on, and separated by, solid carbon material, which prevents nanoparticle sintering and results in a high surface area material. Importantly, the method can be extended to the synthesis of other TMC nanoparticles."

b) "In summary, we report here a facile and scalable two-step process for large-scale production of tunable in size WC nanoparticles."

and c) "Importantly, we envisage applicability of our method to produce other interstitial compound nanoparticles (e.g., transition metal nitrides and phosphides) that could replace noble metal catalysts in a wide range of chemical reactions."

These broad claims are based on data for a single WC material using only SEM/TEM imaging with an EDX map in Figure 2 and Figure S1 along with a PXRD spectrum in Figure S2b. Unfortunately, this single synthetic example and the techniques used are inadequate to confirm metal termination or the broad applicability of the method.

We agree with the reviewer's comment that only one example is inadequate to support the entire methodology we reported in this manuscript. To demonstrate the broader impact of our work and substantiate our claims, we have added more examples of TMC nanoparticles in the SI from other materials and further characterization beyond what was included in the original manuscript. Specifically, we have included (1) STEM tomography analysis of our nano-WC which shows more detailed information about the dispersion of WC nanoparticles on the carbon spheres (Figure 2c), (2) more bright-field TEM images of our catalysts (Figure S3), (3) further discussion

about the synthetic method in the SI, and (4) results on another TMC, namely Mo₂C nanoparticles on carbon spheres (Figure R1 in the SI for reviewers only).

We agree with the reviewer that the confirmation of surface termination of tungsten carbide is important. Unfortunately, there is no efficient pre-characterization method which can provide such information, since WC can easily react with oxygen when exposed to air. *In situ* or *operando* spectroscopy such as XPS, EXAFS and other surface techniques may achieve this target, but to the best of our knowledge, it is impossible to detect the WC surface when it is covered with a micro-meter range polymer. In fact, we are now focusing on applying our WC nanoparticles as the electrode for a fuel cell, and developing *operando* methods to monitor the surface and bulk properties of nano-WC under working conditions using EXAFS and environmental TEM in the National Labs.

The authors should provide at least XPS data, better PXRD patterns, or XAS data to corroborate claims. More specifically, a metal-terminated WC surface will not exhibit a passivation layer with WO_x. Otherwise, it should be denoted as O-terminated. Furthermore, a carbidic C1s shift should be detectable at the surface and it must be comparable or lower in intensity to the adventitious carbon signal. Without this important information, it is unreliable to model a W-terminated material with DFT.

We agree that XPS analysis can offer the carbidic carbon features in the C1s spectrum, but it should be mentioned that our catalyst is not bulk WC which has abundant carbon (amorphous) on the surface. So the huge contribution from the carbon support will significantly influence the identification of carbidic carbon in the WC.

The XRD instrument we used for this study is a Bruker D8 high resolution diffractometer equipped with a LynxEye position sensitive detector, which dramatically increases sensitivity compared with the conventional scintillation detector. The low intensities of the WC patterns shown in Figure S2b (Figure S3b in revised version) indicate the small crystalline size or poor crystallinity of WC nanoparticles. Due to these challenges, we have used electron microscopy to further confirm the small WC nanoparticles.

We have also considered using XAS to characterize nano TMCs. Unfortunately, the bulk information offered by XAS does not distinguish the W from the WC on the surface and inside the carbon sphere (as shown in Fig. S1, W signals are not only from WC on the surface but also from the core of carbon sphere). In addition, XAS cannot differentiate C and O easily, and we have an abundance of O in the carbon sphere as well. Finally, it is not quite straightforward to use XAS to confirm the surface termination (since it is a bulk technique), unless it is a phase pure material.

-If the particles are separated by carbon, it is highly likely they are coated in graphitic coke. Ultimately, it is unclear how the materials are different from those obtained by mixing W precursors with a carbon matrix and heating to carburize the metal with carbon originating from the support, which result in WC nanoparticles encapsulated in graphitic coke. The authors should refer to Z. Yan, M. Cai, and P.K. Shen,

Scientific Reports, 2013, 50, 1646-1653. or R. Ganesan and J.S. Lee, Angew. Chem. Int. Ed. 2005, 44, 6557-6560.

We appreciate that the reviewer pointed out the two papers focusing on nanosized WC synthesis. We would like to emphasize the difference between “carbonization” and “carburization” while comparing the reported methods to ours. We have reported in this paper that a two-step methodology involving hydrothermal carbonization will form solid carbon and a separate W precursor, whereas a temperature-programmed reduction carburization (TPRC in H₂-CH₄ mixture) will form WC particles on the surface of the carbon sphere.

We would also like to point out that both of the cited papers used their prepared tungsten carbide as a **support** for up to **40 wt.% Pt** for the oxygen reduction reaction (ORR) and methanol oxidation. Unfortunately, the activity of tungsten carbide for ORR was missing in the first paper, and **no activity** was observed on the as-prepared W₂C in the second paper.

We have cited these two papers in the revised manuscript. We have also described in the SI experiments (for reviewers only) wherein we annealed our sample collected after the HTC step in a He flow at different temperatures. Unfortunately, the XRD results indicate the formation of WO₃ rather than WC (Figure R2 in the SI for reviewer only).

Furthermore, we have included preliminary results (only for reviewers) of the fuel cell performances using our WC NPs as anode and cathode catalyst (Figure R3 in the SI for reviewers only), which show about 0.09 W/cm² power density. We have also tested our nano-WC as catalysts for both the ORR and the hydrogen oxidation reaction (HOR) (not included in this paper), and they show promising activities. All these additional results support the fact that our WC NPs are able to **catalyze** H₂ oxidation and O₂ reduction.

-The bright white spots on the surfaces of the carbon spheres in Figure 2a are called "well dispersed WC nanoparticles." Based on the scale bar, these particles are at least 20-30 nm in size. Figure S1 shows a cutaway.

We completely understand the reviewer’s concerns and apologize about our sentence in the manuscript. We have modified and added more explanations in the manuscript and SI.

We investigated the morphology of our materials by SEM operated at 3 kV, wherein it is extremely difficult to observe a particle at nm scale. Figure 2a is a low magnification SEM image, the bright white spots are not individual WC particles, but rather an agglomerate of several nanoparticles on the carbon surface. A more detailed image of several WC NPs in close proximity to each other is shown in Figure 2c.

Figure S1 shows a cross-sectional view of the carbon sphere reported in this paper. We use a dual-beam microscope which can locally mill the materials under observation. As shown in Figure S1, the WC nanoparticles only exist on the surface of the carbon sphere; the bulk of the carbon sphere also contains W but not as a WC structure. More details are presented in the SI.

- The authors claim that the method provides size tunability, but no examples are shown where different particles sizes are achieved. Specifically, multiple syntheses have to be performed and the PSDs have to be compared. They must be narrow and statistically different.

We completely agree with the reviewer's comment. To demonstrate the broad applicability of our results, we have included and discussed the synthesis of nano Mo₂C in a document for reviewers only (Figure R1).

-Figure 1 shows nitridations being performed. No data on nitrides is shown. This must be removed. The "Nano Carbide/Nitride" depiction is also inaccurate based on Figure S1. The WC particles are inside the carbon sphere.

We understand the confusion resulting from the omission of TMNs in this manuscript; accordingly, we have now included more discussion in Section 1 of the SI.

-The authors claim to synthesize "hexagonal close packed β -WC". I believe, hexagonal close packed WC is typically referred to as just WC or α -WC. Cubic WC synthesized in a face centered cubic lattice is referred to as β -WC or WC_{1-x}. In Figure 2d, the authors claim to observe the WC(100) plane. β -WC does not have this plane. In Figure S2b, the authors index WC nanoparticles diffractogram to α -WC; however the WC nanoparticle diffractograms cannot be indexed as the intensities are too low.

We thank the reviewer for alerting us to this error. This has been corrected in our revised manuscript.

2. Membrane Testing and Characterization:

The title of the paper includes "Self-hydrating." In the second paragraph of the introduction, the paper is motivated by using nanoparticles incorporated in the membrane to "self-hydrate" the membrane via water produced in the cathode. I agree with the authors that not having to have a complex humidification system would significantly simplify the balance-of-plant. However, if this is a main motivation for the paper, then why was the fuel cell testing performed with both H₂ and O₂ humidifiers maintained at 55 °C? If the authors wish to claim self-hydration and include it in the title, the studies with WC and Pt in the membrane have to be shown without the humidification systems. Otherwise, it is hard to justify the presence of "Self-hydrating" in the title.

We appreciate the reviewer's comment on the "self-hydrating" issue, and apologize for the confusion. We chose "self-hydrating" to describe that our catalyst incorporated within the recast Nafion membrane can catalyze the cross-over gas during fuel cell operation, thereby producing water within the membrane which enhances its proton conductivity. This definition of "self-hydrating" has been used in the fuel cell community, for example, the recent work by Park et al (C.H. Park *et al. Nature*, 532, 2016, 480-483) reports "self-humidifying membranes", although

their fuel cell performance was evaluated at 35% RH which implies the presence of H₂ and O₂ humidifiers.

To address the reviewer's concern, we have performed additional experiments at lower humidity. Comparing the fuel cell performance for the Pt, commercial WC, nano-WC and recast Nafion membranes at low humidity clearly showed that the membranes incorporating the catalyst exhibited improved performance and better proton conductivity. See also response to comment 2 of reviewer 1.

Figure 3a shows that there is no improvement of fuel cell performance with 5 wt.% commercial WC, likely due to the small surface area of WC leading to limited oxidation of crossover H₂. The commercial catalysts are typically prepared through ball-milling, which could lead to some metal termination. Based on Figure S3, the surface area is likely not that low. The authors should run N₂ adsorption/desorption isotherms on the materials and measure the surface area. Based on the authors' hypothesis throughout the paper and the DFT calculations, this commercial WC/Nafion control should have exhibited some improvement, but instead it is actually worse than just Nafion. The data from this control do not support one of the main conclusions of this paper that metal-terminated WC is catalyzing the formation of H₂O from crossover of H₂ to hydrate the membrane.

We understand the reviewer's concern about the specific surface area of WC. N₂ adsorption/desorption is applicable to porous materials. For the non-porous materials in the current study, BET analysis only characterizes the entire surface area of carbon sphere, which is not representative of the actual specific surface area of the WC nanoparticles. More importantly, the presence of catalyst clusters within the membrane matrix can introduce large voids which could impede mass transfer in the through-plane direction and adversely impact its proton conductivity, especially at high humidity. We have presented a more detailed discussion in Section 1 of the SI, that the inclusion of commercial WC within the recast Nafion membrane is detrimental to performance because of its large particle size with low density of active WC species on the surface, as well as the reduction of mass transfer across the membrane's thickness during fuel cell operation. This hypothesis was supported by results from our new impedance measurements and fuel cell evaluations at different humidities which we have added.

3. DFT calculations

The DFT calculations are performed on idealized, metal-terminated WC. While it is understandable that idealized surfaces must be used in DFT calculations, this assumption likely does not hold in the context of the conditions that WC is being subjected to. Specifically, the conditions that the authors are exposing the WC materials to means that they will be passivated in an oxide layer. Not only will WC be passivated, it will likely be thickly passivated. The surface is at least an oxycarbide and more likely is actually similar to WO₃. As a result, it is difficult for me to gain any information from the DFT calculations to understand the observed improvements in membrane stability.

While it is true that in an O₂ environment the WC will become oxidized, it has been shown that negligible oxidation of WC occurs in an *aqueous* environment (Warren, A. *et al. Int. J. of*

Refractory Metals and Hard Materials **1996**, 345-353). Though crossover O₂ is indeed present, crossover H₂, which acts as a reducing agent, is also present. As a result, we expect that within the aqueous fuel cell environment, our WC catalyst should show a lack of oxidation similar to what has been shown previously (Warren, A. *et al. Int. J. of Refractory Metals and Hard Materials* **1996**, 345-353). Therefore, we believe that comparing WC directly to Pt is the most fair and meaningful comparison.

Why was such a low k-point mesh and a low cutoff energy used for the calculations? To model the WC surface under operating conditions, these calculations should be done with at least a 5x5x1 kpoint mesh and a cutoff energy of at least 500 eV.

We note that though a dense *k*-point mesh and a high plane-wave energy cutoff is necessary to converge *total* energies from DFT, energy *differences* (i.e., the reaction energies which we report in this work), in general, converge much faster. However, to ensure that our first-principles calculations were performed at sufficient accuracy, we computed the adsorption energies of H• and OH• on both WC (100) and Pt (111) with a $5 \times 5 \times 1$ *k*-point mesh and 500 eV plane-wave energy cutoff. In all cases, the adsorption energy changed by less than 0.05 eV. Therefore, our methodology is sufficiently accurate. We have added this detail in the Methods section of our manuscript.

Review #3

Several diatomic transition metal carbides or oxides with n valence electrons have similar spectroscopic and chemical properties as transition metal atoms with n d -electrons (Castleman). WC thus simulates Pt and has been investigated as replacement for Pt as a catalyst. In this sense, the present work uses nano-size WC as an additive to Nafion in place of Pt and studies its influence on the power density and the stability of the Nafion fuel cell membrane. An improved behaviour is ascribed to the self-hydrating property due to the conversion of the H₂ and O₂ cross-over gases to water inside the membrane, and DFT calculations find that the H· and HO· radicals adsorb more strongly on WC than on Pt so that they are less available for damaging radical reactions on the polymer electrolyte. Furthermore, a tomographic technique reveals interesting pattern of degradation. This is an interesting topic, and the work combines a variation of different methods. The manuscript reads quite well, and the interpretations sound plausible. However, if one goes into details one gets the impression of a somewhat abridged study with quick conclusions which may be correct but lack certainty in the absence of cross-checks or comparison with literature:

1. Conversion of cross-over gases in the membrane at the added catalyst particles is concluded from the Ohmic region of the polarization curves, but without quoting the membrane resistances derived from the slopes. Perhaps more commonly, the reductions of the open circuit voltages are judged for the same purpose. These, however, seem inconsistent with the suggested interpretation (Figure 3a).

We appreciate the reviewer's comments regarding the OCV plots. We have now added more detailed discussion to the Supporting Information that addresses this point.

Pt/Nafion exhibited the highest OCV (0.888V) due to its higher activity. Commercial WC/Nafion showed the lowest OCV (0.843V) probably due to its lower activity and the presence of voids around the solid particles inclusions within the membrane which may elevate gas crossover. Recast Nafion and nano-WC/Nafion show similar OCVs (0.848V and 0.847V). The OCVs were tested by holding the fuel cell for 1min at OCV after conditioning at 1A/cm². The OCVs always decrease rapidly at the beginning of the test and then reach a balance between water production (beneficial) and gas crossover (harmful) as shown in the durability tests (Figure 3b). Due to these complex and competing effects, we believe that the small difference in OCVs is not an accurate fingerprint of catalyst activity. Rather, we prefer to use the more obvious differences in the ohmic region of the polarization curves to judge catalyst effectiveness.

2. Durability tests are conducted over durations of 100 h OCV. This may be partly justified by the external humidification of the gases which is only 50%, but it is very short compared with normally expected lifetimes of 5000 h under running conditions. What bothers more is the low number of experiments. Is it really enough for a reliable conclusion considering that the break-down that is seen for 5 wt.% Pt/Nafion normally sets in at random time, depending on the actual pinhole situation.

We would like to emphasize that our **accelerated** durability tests were conducted according to the DOE protocol at 90°C and 35% RH.

http://www1.eere.energy.gov/hydrogenandfuelcells/fuelcells/pdfs/component_durability_profile.pdf). This type of aggressive protocol is designed to greatly accelerate the degradation of fuel cell membranes such that membrane lifetime can be conveniently studied in labs (usually in 100 to 300 hours). For example, the baseline Pt/Nafion membrane fails within 100 hours (OCV<0.2V) under this aggressive protocol. Therefore, we conducted all our tests to 100 hours to evaluate durability improvements. Three durability tests were conducted with Pt/Nafion membranes as shown in Figure S7 in SI. All three samples showed very similar trends and failed at about 70 hours. The slight variability in the OCV vs. time profiles is due to the necessarily random formation of pinholes through which reactant gas crosses over, leading to random drops in OCV and eventually, failure. In light of such randomness, the three profiles shown in the figure are quite similar. Hence, we believe that our durability data are reliable.

*3. As a degradation mechanism it is suggested without any reference that H• and HO• radicals in solution react with Nafion via removing SO₃ □ groups. I do not know how H• radicals should form in solution since they are strongly bound to the Pt catalyst (supplementary information). Furthermore, it is known in literature from studies with model compounds that SO₃□ groups are much more inert towards attack of HO• radicals than other sites of Nafion [Dreizler and Roduner, *Fuel Cells*, 12 (2012) 132-140]. Furthermore, with a standard reduction potential of 1.77 V, HO• radicals are strongly oxidizing species [Koppenol and Liebman, *J. Phys. Chem.* 88 (1984) 99.], an aspect which is unfortunately not normally considered in mechanistic work on membrane degradation.*

We apologize for the lack of clarity on this issue. The H• radicals are not formed on the catalyst, but are formed in solution when OH• radicals in solution react with crossover H₂ gas (reaction 2 in the manuscript). Furthermore, there has been evidence of H• contributing to Nafion degradation (Ghassemzadeh, L. *et al. J. Phys. Chem. C* **2010**, 114, 14635-14645; Yu, T. H. *et al. J. Am. Chem. Soc.* **2011**, 133, 19857-19863). Therefore, we cannot exclude degradation of Nafion through H• as a possibility. For chemical degradation of Nafion, the general understanding is that H₂O₂ in PEFC is formed at the cathode from two-electron reduction of O₂ by cross-leakage of O₂ gas. The reactive hydroxyl (•OH) radicals are formed from H₂O₂. (Nosaka, Y, *J. Electrochem. Soc.* **2011**, 158, B430–B433, Ohguri, N, *J. Power Sources* **2010**, 195, 4647–4652 and Gummalla, M, *J. Electrochem. Soc.* **2010**, 157, B1542–B1548.). We agree that the C-S group is more vulnerable to chemical attack from HO• radicals. Therefore, we have revised the manuscript's introduction accordingly.

*4. VASP calculations were limited to the Pt(111) and the WC(100) facets although it is known that oxygen reduction is quite different for the various low-index Pt surfaces, and no comparison is made with literature data on very detailed previous work [Panchenko *et al. J. Electrochem. Soc.* 151 (2004) A2016; and J.K. Norskov *et al. J. Phys. Chem. B*, 108 (2004) 17886.].*

We thank the reviewer for pointing out the reason for choosing to study the (111) surface of Pt and the (100) surface of WC, respectively. Our rationale for selecting these surfaces is that they have the lowest surface energy of the low-index surfaces. Table S1 in the SI provides the computed surface energies for each low-index facet of WC. Furthermore, we compare to a

number of prior works. For example, the surface energy of 3.89 J m^{-2} determined for the metal-terminated WC (100) is comparable to the surface energy of 3.43 J m^{-2} WC (0001) previously reported in the literature (Siegel, D. J. *et al. Surface Science* **2002**, 498, 321-336). For the oxidation of water to an adsorbed OH* species on Pt (111), Nørskov *et al.* reported a reaction energy of +1.05 eV (J.K. Nørskov *et al. J. Phys. Chem. B* **2004**, 108, 17886). Computing this quantity from our data, we predict a reaction energy of +0.99 eV, in close agreement with the results of Nørskov *et al.* By comparison, Panchenko *et al.* determined an adsorption energy of -2.23 eV for hydroxyl on Pt (111) when referencing to the gas phase OH species (Panchenko *et al. J. Electrochem. Soc.* **2004**, 151, A2016). We predict an adsorption energy of -2.29 eV, which is again in close agreement with the published data.

Overall, this work reminds more of a conference proceeding of work in progress, but it does not meet the standard expected for a Nature Comms. Paper.

We thank the reviewer for pointing out shortcomings of the previous version. We believe that the additional experiments and further evidence provided now supports our findings.

Reviewers' comments:

Reviewer #2 (Remarks to the Author):

The authors have submitted a revised version of the original manuscript in order to address several comments provided by the reviewers. In my opinion, there are still many claims that need to be revised for the arguments presented in the manuscript to match the data that is shown. Specifically:

a) In Review 2, Materials Characterization 1, the authors claim that there is no pre-characterization method that can provide confirmation of metal termination of their materials. However, performing ex-situ XPS before the polymer is added to the WC material should allow the authors to probe the surface of the materials. There exist many methods to transfer air sensitive samples onto an XPS. If the authors cannot make the measurements, then the claims regarding the state of the surface of the catalyst should be removed. It will be difficult to deduce from the information presented what is the state of the catalyst surface, especially considering how critical the state of WC is in determining reactivity.

b) In Review 2, Materials Characterization 2, the authors should at least show the W 4f spectrum using XPS.

c) Regarding the comment on "well dispersed WC nanoparticles", the authors now state that not all of the W has been carburized and some fraction does not exist as a WC structure. What percentage of the W is not carburized? What is the efficiency of the process?

d) Regarding the comment on size tunability. The authors' response is both unrelated and insufficient to answer the original comment. The authors claim that their method provides size tunability, but no examples are shown where different particles sizes are achieved. By adding data on Mo₂C the authors fail to show that the size of the WC nanoparticles can be tuned.

e) With respect to the lack of nitridation data, the authors claim that Section 1 of the SI has been updated. I did not find any data related to nitridations for the materials synthesized by the authors. The authors have failed to support their claim that nitrides can be synthesized with this method, so any comments regarding nitrides should be removed.

Reviewer #3 (Remarks to the Author):

The authors provide an extensively revised manuscript and address the reviewer' comments in a competent and substantial way. However, much of the provided discussion and additional information is provided in a supplement "for reviewers only" (see rebuttal letter). This is not appropriate since the publication would be incomplete without the requested information. This information has to be included in the accessible Supplementary Information. Provided that this is done I support publication of the manuscript.

Our response and actions are shown below point by point in red. New revisions in the manuscript and supplement indicated in blue; old revisions in red.

Review #1

The authors have submitted a revised version of the original manuscript in order to address several comments provided by the reviewers. In my opinion, there are still many claims that need to be revised for the arguments presented in the manuscript to match the data that is shown. Specifically:

a) In Review 2, Materials Characterization 1, the authors claim that there is no pre-characterization method that can provide confirmation of metal termination of their materials. However, performing ex-situ XPS before the polymer is added to the WC material should allow the authors to probe the surface of the materials. There exist many methods to transfer air sensitive samples onto an XPS. If the authors cannot make the measurements, then the claims regarding the state of the surface of the catalyst should be removed. It will be difficult to deduce from the information presented what is the state of the catalyst surface, especially considering how critical the state of WC is in determining reactivity.

b) In Review 2, Materials Characterization 2, the authors should at least show the W 4f spectrum using XPS.

In order to address the reviewer's comments a and b, we have performed XPS analysis of the nano-WC and compared it with commercial WC, WO₃ and high temperature annealed (without carburization) nano-WC precursor (nano-WO_x). The XPS results (new Figure S4) show that the WC nanoparticles reported in this paper have a tungsten carbide dominated surface, as evidenced by the characteristic WC (W4f doublet peaks at 31.6 and 33.7 eV) and carbidic carbon (C1s peak at 282.6 eV), the low intensity of oxygen species (O1s), and the metallic nature with high density at the Fermi level (valence band analysis). Together with our previous intensive characterization results, we confirm that the two-step process reported in this study can efficiently produce large-scale, well-dispersed WC nanoparticles.

c) Regarding the comment on "well dispersed WC nanoparticles", the authors now state that not all of the W has been carburized and some fraction does not exist as a WC structure. What percentage of the W is not carburized? What is the efficiency of the process?

We appreciate the reviewer bringing up this valuable issue of estimating the efficiency of our synthetic protocol. We have employed TGA analysis of our nano-WC catalyst under flowing air to quantify the total amount of W which is around 60 wt.%. According to the surface composition analysis by XPS, the nano-WC has about 52 wt.% tungsten on the top layers of WC and carbon sphere. The results included in the Supplementary Information (Figure S4 and S6) show that more than 80% of W can be carburized through our synthetic method.

d) Regarding the comment on size tunability. The authors' response is both unrelated and insufficient to answer the original comment. The authors claim that their method provides size tunability, but no

examples are shown where different particles sizes are achieved. By adding data on Mo₂C the authors fail to show that the size of the WC nanoparticles can be tuned.

We agree with reviewer's comment on this issue. For this study, it is more important to provide a stable and highly dispersed tungsten carbide for catalytic reaction rather than investigating the particle size effect. Thus we decided to exclude the claim of "size tunability".

e) With respect to the lack of nitridation data, the authors claim that Section 1 of the SI has been updated. I did not find any data related to nitridations for the materials synthesized by the authors. The authors have failed to support their claim that nitrides can be synthesized with this method, so any comments regarding nitrides should be removed.

We appreciate the reviewer's comment on this issue and apologize for any confusion. Based also on reviewer 3, we provide now in the supplement data for Mo₂C as another example of the generality of our approach. Given the consensus that early transition metal nitrides can be formed given similar paths to carbides (S. Ted Oyama, Transition Metal Carbides, Nitrides, and Phosphides, Handbook of Heterogeneous Catalysis, WILEY-VCH) and a large number of papers reporting that transition metal carbides can be transformed to nitrides by ammonolysis (thermal treatment with ammonia), we believe that this should be straightforward but is something we have not done, as it was outside the objective of our work. We have revised our manuscript to make this clear.

Review #3

The authors provide an extensively revised manuscript and address the reviewer' comments in a competent and substantial way. However, much of the provided discussion and additional information is provided in a supplement "for reviewers only" (see rebuttal letter). This is not appropriate since the publication would be incomplete without the requested information. This information has to be included in the accessible Supplementary Information. Provided that this is done I support publication of the manuscript.

We appreciate the reviewer's effort and comment to revise this work. We have included all experimental results into the main manuscript and supplementary information together with additional characterization results.

REVIEWERS' COMMENTS:

Reviewer #2 (Remarks to the Author):

The authors addressed the main comments by the reviewers. Focusing the manuscript to tungsten carbide, while removing some of the broader claims has helped strengthen the manuscript. While , in my opinion, there are still discrepancies between the general claims and the actual results presented in this study, the article should be suitable for publication.